# Studying Plant–Insect Interactions through the Analyses of the Diversity, Composition, and Functional Inference of Their Bacteriomes

**DOI:** 10.3390/microorganisms11010040

**Published:** 2022-12-22

**Authors:** Zyanya Mayoral-Peña, Víctor Lázaro-Vidal, Juan Fornoni, Roberto Álvarez-Martínez, Etzel Garrido

**Affiliations:** 1Unidad de Microbiología Básica y Aplicada, Facultad de Ciencias Naturales, Universidad Autónoma de Querétaro, Santiago de Querétaro 76123, Mexico; 2Departamento de Ecología Evolutiva, Instituto de Ecología, Universidad Nacional Autónoma de México, Ciudad de México 04510, Mexico

**Keywords:** bacteriomes, co-abundance networks, diversity, foliar microbiota, functional inference, gut microbiota, plant–insect interaction

## Abstract

As with many other trophic interactions, the interchange of microorganisms between plants and their herbivorous insects is unavoidable. To test the hypothesis that the composition and diversity of the insect bacteriome are driven by the bacteriome of the plant, the bacteriomes of both the plant *Datura inoxia* and its specialist insect *Lema daturaphila* were characterised using 16S sRNA gene amplicon sequencing. Specifically, the bacteriomes associated with seeds, leaves, eggs, guts, and frass were described and compared. Then, the functions of the most abundant bacterial lineages found in the samples were inferred. Finally, the patterns of co-abundance among both bacteriomes were determined following a multilayer network approach. In accordance with our hypothesis, most genera were shared between plants and insects, but their abundances differed significantly within the samples collected. In the insect tissues, the most abundant genera were *Pseudomonas* (24.64%) in the eggs, *Serratia* (88.46%) in the gut, and *Pseudomonas* (36.27%) in the frass. In contrast, the most abundant ones in the plant were *Serratia* (40%) in seeds, *Serratia* (67%) in foliar endophytes, and *Hymenobacter* (12.85%) in foliar epiphytes. Indeed, PERMANOVA analysis showed that the composition of the bacteriomes was clustered by sample type (*F* = 9.36, *p* < 0.001). Functional inferences relevant to the interaction showed that in the plant samples, the category of Biosynthesis of secondary metabolites was significantly abundant (1.4%). In turn, the category of Xenobiotics degradation and metabolism was significantly present (2.5%) in the insect samples. Finally, the phyla Proteobacteria and Actinobacteriota showed a pattern of co-abundance in the insect but not in the plant, suggesting that the co-abundance and not the presence–absence patterns might be more important when studying ecological interactions.

## 1. Introduction

Plants and their herbivorous insects have always interacted under a microbial milieu that allows the constant and reciprocal exchange of all the microorganisms associated with both interacting species [1,2,3,4]. Because the presence of all these microorganisms is not innocuous, their effects extend to the ecology and evolution of their hosts as well as their ecological interactions [1]. Indeed, over the past years, excellent reviews about the ecological and evolutionary implications of microorganisms on plant–insect interactions have been published [5,6,7,8]. However, most of the studies to date have focused on describing the microbiota within individual multicellular hosts, either plants or insects. They have not yet identified the functional role of most microbial lineages. Here, we aim to expand our understanding of plant–insect interactions by simultaneously studying their bacteriomes to assess possible patterns of co-abundances among bacterial strains belonging to the core microbiome of both organisms that might be important in the interaction.

Overall, the microbiota refers to all the microorganisms associated with a given host, and it is composed of different taxonomic groups such as bacteria, fungi, algae, viruses, archaea, and, less frequently, nematodes and protozoa [9,10]. However, in both plants and insect herbivores, the most abundant group is bacteria [10,11], thus the term bacteriome is preferred when characterising only the bacterial groups [12]. Those microorganisms associated with the aerial parts of a plant constitute the phyllosphere which can be further classified into the endosphere (microbes residing within the tissue) and the phylloplane (microbes residing on the surface) [13,14]. In turn, the microorganisms that reside within the tissues of an insect are called endogenous, while those on the surface of their bodies are exogenous microbes [15]. Recent studies have described the microbiota associated with both plants and insects, but when evaluating its potential effects, the same studies have only focused on one of the interacting species [16,17,18]. That is, the role of the microorganisms has been traditionally studied in terms of its effects on the survival and adaptation of each host but not in terms of understanding the ecological interaction between the hosts. If reciprocal effects between the foliar and gut bacteriomes have the potential to shape plant–insect interactions [19,20,21,22], a new player must be considered as part of the co-evolutionary arena. Thus, studying possible associations and interactions among the bacteriomes of both plants and insects might help us understand the ecology and evolution of the interaction.

The phyllosphere is, at some point, inevitably consumed by phytophagous insects. Recent evidence suggests that phyllosphere bacteria can indeed colonise the insect gut [23,24,25,26], become part of the transient gut microbiota [25,27] and affect insect survival and performance [19,28,29,30] by facilitating detoxification and nutrient acquisition [10,20,31]. Most of the transient gut microbiota come from the foliar tissue consumed and is usually excreted in the faeces [32]. Thus, the composition of the intestinal microbiota can be affected by the food source, the host genotype, the composition of the egg microbiota, and the biogeographic region it inhabits [24,33,34,35]. Indeed, it has been suggested that diet plays a relevant role in the composition of the gut microbiota of various insects [16,17,36,37]. In turn, it has been shown that the gut microbiota affects the pattern and intensity of foliar damage [2,38] as well as the diversity and composition of the phyllosphere [39,40] via the incorporation of some insect-associated microbes by the plant [41,42]. Moreover, the gut bacteriome can alter the expression of putative mechanisms of plant defence [43,44,45]. It has been shown that the microbiota associated with the oral secretions of the insects reduces the levels of induced plant defences, favouring food intake [1,2]. Therefore, microorganisms can alter the expression of plants and insect traits involved in their reciprocal evolutionary responses. After consumption, insect frass might come into contact with the surface of the leaves, and it has been shown that certain bacterial strains can be effectively transferred to the surface of neighbouring leaves or plants [46]. Whether this kind of phyllosphere transfer is possible throughout frass contact remains to be tested (but see [47]).

The aim of the study was threefold. First, to describe and compare the diversity and composition of the bacteriome associated with the seeds and leaves of the host plant *Datura inoxia* and the bacteriome associated with the eggs, gut, and frass of its specialist insect, *Lema daturaphila*. Given the nature of the interaction, we hypothesize a high degree of overlap in the composition of both sets of bacteriomes because the foliar tissue—along with all the microbes associated with it—is necessarily consumed by the larvae. However, differences in the abundance of those shared bacterial strains are expected. Second, to determine the functional inferences of the most abundant bacterial families. Finally, to assess the patterns of co-abundance and analyse the spatial dynamic of both sets of bacteriomes following a multilayer network approach. To our knowledge, this is the first attempt to characterize the diversity, the patterns of co-abundance, and the spatial dynamics of the bacteriomes associated with both hosts to shed light into their ecological interaction.

## 2. Materials and Methods

### 2.1. Study System and Sample Collection

*Lema daturaphila* Kogan & Goeden (Coleoptera: Chrysomelidae) is the specialist herbivorous insect of *Datura inoxia* L., an annual plant belonging to the Solanaceae. Under natural conditions, *D. inoxia* typically grows in disturbed areas, forming uneven patches, and during the peak season, an individual plant can experience more than 80% of foliar loss due to the presence of *L. daturaphila* (personal observation). Both the eggs and larvae of the beetle can be parasitised. In the eggs, the presence of a specialist wasp has been reported [48], and the larvae of all stages can be parasitized by either a generalist fly (Diptera: Tachinidae) or a specialist wasp (Hymenoptera: Ichneumonidae) [49]. All the biological samples used in this study were collected from a natural population in the municipality of El Marqués, Querétaro, México (20.66° N, 100.32° W). During the months of August and September of 2019, the following samples were collected: ten undamaged leaves and five unripe fruits from each of four healthy plants, as well as all the clusters of eggs and larvae of all developmental stages found in the same plants and neighbouring plants. All the samples were kept on ice and taken to the laboratory to be processed.

In a flux cabinet, epiphytes from seeds and leaves were recovered with a phyllosphere removal buffer (6.75 g of KH2PO4, 8.75 g of K2HPO4, and 1 mL of Triton X-100 per litre, 1 min) and a 0.9% NaCl solution (1 min). Phyllosphere pellets were then obtained by centrifugation and stored at −80 °C prior to DNA extraction. To obtain endophytes, the samples were then surface-sterilised with serial washes: 70% ethanol (2 min), 10% sodium hypochlorite (1 min), and four final washes with sterile water (2 min each) (modified from [50]). To validate our epiphyte-removal procedure, the leaves and seeds were then imprinted in Petri dishes filled with TSA medium. The dishes were incubated at 28 °C and checked for one month for any microbial growth. Finally, seed coats were manually removed, and approximately 270 mg of uncoated seeds and 580 mg of leaves per plant (from our four biological replicates each) were stored at −80 °C prior to DNA extraction.

To assure that all the insect samples were not parasitised, a colony was initiated with the eggs and larvae collected from the field and maintained under laboratory conditions (12L:12D photoperiod, 25 °C, fed with plants from the same population, without the presence of parasitoids or pathogens). Once there were plenty of eggs in our colony (late October), we collected four biological samples, each consisting of three to four clutches (approximately fifty eggs in total). To remove the exogenous microbes, the eggs were placed in 1.5 mL tubes and washed with Microdacyn (OCULUS, Sodium < 55 ppm, and Chlorine < 80 ppm) for 15 min and with sterile water for 30 min. Prior to dissecting the larvae, the exogenous microbes were removed following the same procedure as the one for epiphytes removal. Only third- and fourth-instar larvae were dissected to obtain their entire guts. All dissections were performed in a flux cabinet. Four biological samples were taken, each consisting of around ten to fifteen guts. Finally, during October, all the frass produced by fourth-instar larvae of our colony was collected daily with the use of a brush, and four biological samples consisting of approximately 250–300 mg were kept at −80 °C until DNA extraction.

### 2.2. DNA Extraction and Generation of 16S rRNA Amplicons

Epiphytes from seeds and leaves were centrifuged at 10,000 rpm for 3 min, the supernatant was removed, and the resultant pellet was diluted to a final volume of 2 mL. Genomic DNA was then extracted using a DNeasy blood and tissue kit (QIAGEN, Germantown, MD, USA) following the manufacturer’s instructions. Endophyte samples were first macerated in liquid nitrogen, and genomic DNA was extracted using a DNeasy PowerSoil kit (QIAGEN, Germantown, MD) following the manufacturer’s instructions. Surface-sterilised eggs and entire guts were macerated in liquid nitrogen. DNA extraction from eggs was done following the protocol described by [51]. Genomic DNA from guts was extracted using a DNeasy blood and tissue kit (QIAGEN, Germantown, MD) following the manufacturer’s instructions. Finally, extraction of genomic DNA from frass samples was done with slight modifications of the IHMS procedure described by [52]. For all the samples, the v3-v4 hypervariable region of the 16S rRNA gene was sequenced in a 2 × 250 bp paired-end run using the Illumina Miseq platform. Sequencing was carried out at the Instituto Nacional de Medicina Genómica (INMEGEN), México.

### 2.3. Data Analyses

All statistical and bioinformatics analyses, as well as all plots were done using R and its libraries [53]. Raw sequencing data were processed following the DADA2 pipeline [54]. Because of the low Phred quality, it was decided to work only with the forward reads. The Phred quality threshold was ≥20, and the length of the sequences was 230 pb. The Silva 16S rRNA (release 138 [55,56]) database was used for taxonomic identification. The diversity of the different samples was estimated using the Rènyi profile implemented in the vegan v. 2.5–7 package [57], and the results were plotted using ggplot2 v. 3.3.2 [58]. The Rènyi diversity profile summarises various aspects of alpha diversity such as richness, dominance, and equity [59]. When calculating the Rènyi profile, if the value of alpha tends to 1, 2, and infinity, it behaves like the Shannon–Weaver diversity index, the logarithm of the reciprocal Simpson diversity index, and the Berger–Parker diversity index, respectively. Thus, if a Rènyi profile of a particular sample is consistently higher, it is considered as the most diverse, and those profiles that tend to be horizontal suggest that the species are distributed with less equity. From the Rènyi profiles, the Shannon diversity index and richness values of the different samples were then compared with an ANOVA test. The phylogenetic diversity of the samples was estimated using the picante v. 1.8.2 package [60]. To visualise possible dissimilarities among bacterial communities, an RDA was performed using the Bray–Curtis distances [61]. For this, a logarithmic transformation of the data was first made, and a PERMANOVA analysis was then carried out. To test for significant differences in the abundances of the bacteria present in the samples, a Wald’s parametric test was performed using the DEseq2 function from the phyloseq package v. 1.32.0 [62]. Considering those ASVs with abundances greater than 10% in the samples processed, the functional inference analysis was done using the tax4fun2 package v. 1.1.5 [63]. Differences in the functional categories among samples were tested with an ANOVA and a Kruskal–Wallis test.

Finally, the co-abundance networks were inferred and built following a multilayer network approach using the igraph [64] and MuxViz [65] R-packages. In a multilayer network, co-abundances are represented by edges connecting nodes, which represent the individual species in the community. The thickness of the edges represents the strength of the co-abundance relationship between the species, with thicker edges indicating a stronger relationship. This type of network can be useful for visualising the relationships between species in a microbial community and for identifying potential interactions or correlations between them. Overall, studying co-abundances in a multilayer network can provide valuable insights into the complex interactions within a microbial community. Both networks were inferred from the abundance tables using SparCC and SPIEC-EASI from the SpiecEasi R-package [66]. SparCC (Sparse Correlations for Compositional data) estimates the linear Pearson correlations between the log-transformed components of the abundance table, represented as nodes in the network. The algorithm does not assume that the data must have a normal distribution and is very robust about the assumption that the interconnectivity is sparse [67]. On the other hand, SPIEC-EASI (SParse InversE Covariance Estimation for Ecological Association Inference) estimates the interaction network by neighbourhood selection or by sparse inverse covariance selection, assuming that interconnectivity between nodes is low. This method avoids the detection of indirect correlations [66]. The networks were coloured at the phylum level before adding them to the multilayer networks using the MuxViz package.

## 3. Results

### 3.1. Composition and Diversity of the Bacteriomes

From 21 samples, 84,444 high-quality and filtered read sequences and 3021 ASVs were analysed, with a median of 3452 reads and 146 ASVs per sample (Appendix A). The following taxonomic categories were identified from the ASV sequences: 100% to phylum, 99.37% to class, 94.25% to order, 88.91% to family, and 69.98% to genus. After removing chloroplasts and mitochondria, the number of ASVs identified in the seed and foliar endophyte samples were low (Appendix A). Thus, these samples were not included in some of the analyses, mainly in those comparing the diversity and abundance of the bacterial communities.

According to the Rènyi profile, foliar epiphytes had the highest diversity of all the bacterial communities analysed (Figure 1A), while the communities associated with the gut and frass had ASVs that are considered dominant (Figure 1A). For the alpha diversity, the ranges observed for total ASVs were 100–294, for the Shannon index were 3.02–5.09, and for the Faith PD index were 1.89–10.24 (Figure 1B–D). The richness (*p* < 0.0001) and diversity (*p* = 0.0017) of the gut samples were significantly lower compared with those of the other samples (Figure 1B). The bacterial community associated with the episphere had the highest Faith PD index and the highest number of taxa but in lower abundance (*p* < 0.0001) (Figure 1D). In contrast, the gut and frass samples had the lowest diversity (Figure 1D), with the genus *Serratia* dominating in the gut samples and *Pseudomonas* dominating in the frass samples (Appendix A).

The most abundant phyla identified in the plant samples were Proteobacteria (71.1%), Actinobacteriota (17.3%), Firmicutes (0.44%), and Bacteroidota (6.57%). At the genus level, eight, eighty-five, and six genera were identified in the seed endophytes, foliar epi-, and endophyte, respectively. Specifically, the most abundant genera in the seed endophyte samples were Serratia (40%), Pantoea (27.9%), and Pseudomonas/Streptomonas (20%). As for the foliar epiphytes, the most abundant genera were Hymenobacter (12.85%), Pseudomonas (6.49%), and Sphingomonas (7.65%), while in the foliar endophytes they were Serratia (67%), Escherichia/Shigella (50%), Janthinobacterium (50%), Stenotrophomonas (50%), and Microvirga (50%) (Figure 2). On the other hand, the most abundant phyla detected in the insect samples were Proteobacteria (92.79%), Actinobacteriota (3.11%), Firmicutes (2.43%), and Bacteroidota (1.34%). From the total of 77 genera identified, only the twelve most abundant represented 70.2% of the total readings (Figure 3). Seventy-one, 56, and 24 genera were identified in the eggs, gut, and frass, respectively. The most abundant genera in the eggs were Pseudomonas (24.64%), Escherichia/Shigella (18.59%), and Stenotrophomonas (4.08%). In the gut, the most abundant were Serratia (88.46%), Escherichia/Shigella (15.69%), and Kosakonia (0.36%). In the frass, Pseudomonas (36.27%), Pantoea (10.22%) and Serratia (7.18%) were the most abundant ones (Figure 2).

Overall, the PCA showed that the bacteriomes were grouped by type of sample (Figure 3), and significant differences were detected among them via a PERMANOVA (*F* = 9.36, *p* < 0.001). Interestingly, the endophytes from seeds and leaves overlapped (Figure 3). Moreover, the gut samples were closer to both endophytes from the seeds and leaves than from any other insect sample, while frass samples were the most distant from the rest (Figure 3). Further comparisons among bacteriomes showed ASVs and genera shared between the different samples (Appendix A). Specifically, genera like *Azomonas*, *Bacillus*, *Brucella*, *Serratia*, *Pantoea*, *Pseudomonas*, *Roseomonas*, *Lawsonella*, *Microvirga*, and *Enterobacter* were present in the foliar epiphytes as well as in the eggs and gut, albeit in different abundances (Appendix A). Several genera like *Azomonas*, *Enterobacter*, *Kosakonia*, *Massilia*, *Pantoea*, *Pseudomonas*, *Serratia*, *Siccibacter*, and *Stenotrophomonas* were shared between the epiphytes, gut, and frass. In contrast, *Methylobacterium-Methylorubrum*, *Bacillus*, *Hymenobacter*, *Escherichia-Shigella*, *Sphingomonas*, *Skermanella*, *Janthinobacterium*, and *Lechevalieria* were shared between the epiphytes and gut but were absent from the frass samples (Appendix A).

### 3.2. Functional Inference of the ASVs Found in the Samples

In both plant and insect samples, 330 KEGG Orthology groups (KOs) at level 1 were identified. From the 41 categories at level 2 of KEGG, only 21 were related to plant–insect interactions, and the rest were classified as “Others”. Those 21 categories grouped at level 2 belonged to four categories at level 3 which were Metabolism, Cellular process, Genetic information processing, and Environmental information processing. However, most of the predicted functions belonged to the Metabolism category, suggesting that one of the main functions of the microorganisms associated is related to the metabolism of their hosts. Specifically, the most abundant categories (mean ± sd) at level 2 were Global and overview maps (eggs: 35.21 ± 0.03; gut: 35.38 ± 0.81; frass: 34.8 ± 0.12; seeds: 36.87 ± 3.36; foliar endophytes: 34.24 ± 2.03; foliar epiphytes: 36.35 ± 0.27); Carbohydrate metabolism (eggs: 9.6 ± 0.46; gut: 10.47 ± 0.45; frass: 9.1 ± 0.27; seeds: 10.14 ± 1.33; foliar endophytes: 9.72 ± 1.37; foliar epiphytes: 9.09 ± 0.13), and Amino acid metabolism (eggs: 9.2 ± 0.38; gut: 8.38 ± 0.9; frass: 9 ± 0.19; seeds: 8.83 ± 1.08; foliar endophytes: 8.49 ± 0.44; foliar epiphytes: 9.14 ± 0.07). On contrary, the least abundant categories were Transcription (eggs: 0.063 ± 0.002; gut: 0.067 ± 0.008; frass: 0.06 ± 0.001; seeds: 0.07 ± 0.008; foliar endophytes: 0.06 ± 0.014; foliar epiphytes: 0.069 ± 0.0006) and Transport and catabolism (eggs: 0.24 ± 0.020; gut: 0.19 ± 0.03; frass: 0.20 ± 0.016; seeds: 0.17 ± 0.05; foliar endophytes: 0.22 ± 0.004; foliar epiphytes: 0.3 ± 0.01) (Figure 4). The functions of Cellular process, Genetic information processing, and Environmental information processing are mainly related to the survival and development of the bacteria. In contrast, the Metabolism category also includes functions that could be relevant to the plant–insect interaction such as the synthesis of secondary metabolites, enzymes related to carbohydrate metabolism, and the degradation of xenobiotics.

Interestingly, the level 2 category of Metabolism of terpenoids and polyketides had the highest abundance in foliar epiphytes relative to all other samples but eggs (Kruskal–Wallis, *p* = 0.00004). At level 1 specifically, the most abundant categories in the epiphytes were Sesquiterpenoid and triterpenoid biosynthesis (ANOVA, *p* > 0.01; Figure 5A), Biosynthesis of terpenoids and steroids (ANOVA, *p* > 0.01; Figure 5B), Monoterpenoid biosynthesis (ANOVA, *p* > 0.001; Figure 5C), and Indole alkaloid biosynthesis (ANOVA, *p >* 0.01; Figure 5D). In contrast, in the samples of eggs, gut, and frass, the level 1 category Drug metabolism—other enzymes had higher abundance than the category Drug metabolism—cytochrome p450 (*t*-test, *p* > 0.05; Figure 5F).

### 3.3. Co-Abundance Networks

Following a multilayer network approach, the co-abundance networks for both the plant and the insect were also inferred (Figure 6). The network connections correspond to the co-abundance patterns. That is, if two nodes are connected, they have similar abundance profiles in the different samples, and the thickness of the connection lines indicates the similarity of these abundances. Building a multilayer network allow us to simultaneously visualise how the correlations in abundance changes among the bacterial strains associated with both interacting hosts. In any insect–plant interaction where the microbiota of the insects is evidently affected by the microbiota of the leaves they are consuming, multilayer networks help us detect patterns and changes in the co-abundance profiles of the bacteria associated with both hosts. In the samples processed, Proteobacteria was the most abundant and most connected phyla in both networks (Figure 6, red nodes). The second most abundant phyla were Actinobacteria and Firmicutes. Both phyla belonged to the core cluster, showing that the presence of both phyla was correlated. In other words, the abundance of one phylum affects the abundance of the other and vice versa. Surprisingly, we found that in the insect co-abundance network, the connection pattern between Proteobacteria and Actinobacteriota was common, but in the plant co-abundance network, this pattern was not evident. These differences in connectivity within the networks could be explained because communities associated with the fecal matter of the insect *L. daturaphila* (mainly Proteobacteria, red nodes in Figure 6) were found in both networks inferred from the samples. In addition, nitrogen-fixing bacterial communities (Cyanobacteria, blue nodes in Figure 6) were found to be unique to the plant tissues sampled, which suggest a mutualistic interaction. However, some connections with the Cyanobacteria phylum were absent or there was no direct connection (i.e., co-abundance pattern) in the plant network as was the case between Chloroflexi, Cyanobacteria, and Proteobacteria (top three nodes in both networks), suggesting that the abundances of those other phyla could affect the dynamic within the bacterial communities associated with the plant *D. inoxia*.

## 4. Discussion

### 4.1. The Composition and Diversity of the Plant and Insect Bacteriomes

The foliar epiphytes of *D. inoxia* had the highest Richness and Faith PD indices, probably because of the continuous exposure to microorganisms associated with soil, air, and water [68]. Among the most abundant genera in the phyllosphere were Hymenobacter and Sphingomonas, which are common in urban patches [69]. Other genera like *Pseudomonas*, *Methylobacterium*, *Massilia*, *Aureimonas*, *Listeria*, *Staphylococcus*, and *Pantoea* had been previously reported as part of the phyllosphere of plants belonging to the Solanaceae family, to which *D. inoxia* also belongs [70,71,72]. Interestingly, the phyllosphere had the highest value in phylogenetic diversity, but the abundances of its ASVs were low. It has been shown that interactions among different bacterial strains already colonising the leaves affect the establishment of new strains [15], thus the incorporation of new microorganisms into the phyllosphere is not stochastic. Moreover, the oligotrophic environment of the leaf surface [73] might also alter the composition and abundance of the phyllosphere.

The egg bacteriome of *L. daturaphila* was more diverse than the one from the gut, probably because the gut environment is more selective [74,75]. However, in other species such as *Copris incertus* and *Brithys crini*, no significant differences are observed in the bacterial diversity associated with both eggs and gut [76,77]. The gut bacteriome was the least diverse and the one with the greatest variation between samples. Other studies have found that the environment and the soil microbiome affect the composition of the gut bacteriome [16,31,78]. Thus, it is possible that variation in the foliar tissue used to maintain our insect colony was the source for such variation among gut samples [3,21,42,79]. Differences in the composition of the gut bacteriome have important ecological implications, given their direct effect on insect survival and performance [16,19,23,42,80,81]. Finally, the phylogenetic diversity was the lowest in the frass samples, given that *Pseudomonas* was the most dominant genus. It has been suggested that the frass microbiota must reflect the intestinal one [82]. In this study, however, this pattern was not found, but significant differences were detected not only in the composition but also in the abundance of both bacteriomes, as other studies have shown [79]. The presence of *Bacillus* in the frass bacteriome is interesting because it has been shown to promote the production of volatile alkyl disulphides that act as attractants [20].

When comparing the composition and abundances of the plant and insect bacteriomes, the egg bacteriome resembled that of the foliar epiphytes, suggesting that, as a result of the interaction, some microorganisms associated with the plant have the potential to reach the germ cells of the insect, ensuring their permanence and incorporation [20]. Among the genera present in both plants and insects were *Serratia* and *Pseudomonas*. Interestingly, the presence of these genera in plants has been shown to affect their development [3,83], while in insects, they can be obligate symbionts of entomopathogens [84,85,86,87]. Moreover, *Serratia* is also known to promote an anaerobic environment in the gut, which could favour the presence of specific strains over others [88]. While Enterobacter, Stenotrophomonas, Pseudomonas, and Sphingomonas were also present in both plant and insect tissues, they were all more abundant in the plant, where they are known to affect the expression of plant defences [42] and protect against pathogens [70,71,72,73]. It is possible that the presence of all these genera in the insect tissues is the result of their direct interaction with their host plant. Whether these microorganisms affect insect performance remains to be tested.

### 4.2. Functional Inferences of the ASVs Found in the Samples

The functional inferences obtained from the samples collected indicate that the bacteriomes of the plant *D. inoxia* and its specialist insect *L. daturaphila* mainly contribute to their host’s metabolisms, as has been reported previously for other plants [89,90] and insects [76,91]. In particular, the bacteriome associated with the insect samples collected mainly contributed to the Carbohydrate metabolism function, which is relevant because foliar consumption involves the degradation of polysaccharides, a process in which the gut microbiota is known to participate [92,93]. Functions related with the metabolism of starch and sucrose were also detected, which again are relevant in the interaction because both polymers are the main storage products in leaves [94]. In addition, β-glucosidase enzymes were detected, which has been linked to the degradation of cellulose [95] and has been detected in the bacteriomes of other insects [75,76,77,78,79,80,81,82,83,84,85,86,87,88,89,90,91,92,93,94,95]. These enzymes were also detected in the epiphytes, which protect plants against herbivores [96]. In addition to carbohydrate metabolism, functions related to the metabolism of different amino acids like alanine, tyrosine, glutathione, glutamine, and glutamate were detected [90,91,97,98,99]. Specifically, the bacteriomes contribute to the metabolism of cofactors, vitamins, and lipids, thus contributing to the number of nutrients in their hosts. Given that the vegetative tissue does not always provide herbivores with the necessary nutrients, it has been proposed that the gut microbiota might contribute to the efficiency of food consumption [91,93] with possible benefits for the insect.

The bacteriome associated with the plant tissues participates in the production of secondary metabolites as well as proteases which all reduce the amount of damage caused by herbivores [8,100]. The biosynthesis of secondary metabolites such as monoterpenes, sesquiterpenes, and triterpenes was higher in epiphytes. In other studies, the potential of epiphytes in the production of secondary metabolites was also reported [100,101]. In addition to the terpenes, the biosynthesis of alkaloids was found, among which was the biosynthesis of an indole alkaloid and tropane in both the plant and the insect. Although tropane was not significantly higher in the plant than in the insect, it is known that tropane is one of the primary secondary metabolites detected in *D. inoxia*. In *D. stramonium*, it is known to participate in the resistance mechanisms against herbivores [102,103,104]. In addition to secondary metabolites, the presence of enzymes such as inhibitors of cysteine peptidase, which protect plants from some insect larvae [105], was inferred. Those enzymes were mainly seen in plant epiphytes. Therefore, the bacteriome associated with the epiphytes could contribute to the resistance mechanisms of plants. In this sense, the magnitude of the impact of some epiphytic metabolites on the survival or performance of *L. daturaphila* could be empirically evaluated in the future, given that as part of the arms race and local adaptation patterns, the bacteriome associated with the insect might improve its detoxification mechanisms [5]. In turn, it would be interesting to evaluate the quantitative relevance provided by the bacteriome in the production of secondary metabolites. Because the amounts produced by the foliar bacteriome are probably low and might not contribute to host performance, it is still possible that the production of these secondary metabolites promotes the expression of plant genes related to the production of secondary metabolites.

Functions relevant in the interaction, like those related to the degradation of xenobiotics [76,91,106,109], were also found. In this category, some enzymes that were detected in the epiphytes as well as in the insect were cytochrome p450, dye-decolourizing peroxidase (DyP), and glutathione-S-transferase (GST), all of which have been previously reported in plants [110,111] and insects [91,112]. In plants, cytochrome p450 enzymes regulate the synthesis of allelochemicals [110]. The microbial activity of this enzyme could increase plant defence because allelochemicals affect the assimilation of nutrients [113,114]. It is also possible that the epiphytic community, which also participates in the synthesis of this enzyme, might contribute to the detoxification mechanisms in the insect [21,107]. Other enzymes that could be involved are the dye-decolourizing peroxidase (DyP), glutathione-S-transferase (GST), and phenol oxidases that participate in the degradation of toluene or xylene [91]. One of the many functions attributed to the gut microbiota is the degradation of xenobiotics [3,25,91,93], thus, the efficiency of detoxification mechanisms might be better understood as the result of the interaction between these enzymes in both the plant and the insect [92,108,115,116].

### 4.3. Co-Abundance Networks

In the multilayer networks built for this study, one layer corresponds to the co-abundance network among bacterial strains present in all the tissues of the insect, and the second layer shows the same kind of network but in the plant. In the last decade, we have acknowledged that multilayer networks represent a powerful tool to study a large variety of ecological systems like plant–pollinator or plant–microbe interactions [117], but, to our knowledge, no study to date has built multilayer networks to study plant–insect interactions. In the case for the interaction between the plant *D. inoxia* and its specialist insect *L. daturaphila*, most phyla and genera were present in both the plant and the insect samples processed, but their abundances and connectivity within the networks were different. In other words, the abundance of certain phyla was correlated in the insect but not in the plant. Whether this differential co-abundance pattern has consequences in terms of the ecological interactions among the bacterial strains remains unknown.

Specifically, in the plant network, clusters of bacteria formed by the Chloroflexi, Cyanobacteria, and Proteobacteria phyla were associated with functions like nitrogen fixation. These bacterial communities, however, were absent in the insect tissues, indicating that these bacterial strains might be plant-specific (Figure 6). Moreover, these phyla were correlated within the plant co-abundance network, suggesting their importance in establishing mutualistic interactions with the plant. The nitrogen fixation by bacterial species such as *Azotobacter*, *Clostridium*, or *Rhizobium* has been reported to be of vital importance in plant growth [118]. Some of the genera found exclusively in the plant tissues like *Pseudomonas* [119], *Pantoea* [120], and *Siccibacter* [121] have been associated with nitrogen fixation and auxin biosynthesis [122]. It should be noted that rhizobacteria such as *Bacillus amyloliquefaciens* and *Pseudomonas putida* can promote the activation of the jasmonic acid and salicylate acid pathway, which increases the resistance levels to insects such as caterpillars [123]. Meanwhile, genera such as *Serratia*, *Shigella*, and *Escherichia* were more abundant in the *L. daturaphila* layer (Figure 6), although they also had an important presence in the superficial tissues of the plant. That is, these genera were found in both layers, but they were especially abundant in the gut samples. The fact that these communities were found in both hosts could be related to the fact that, when feeding on the plant, the larvae defecate on its surface, also leaving traces of these bacterial populations. It has been reported that the larvae of *L. daturaphila* tend to cover themselves with their own frass while remaining on the plant they feed on, which could be a defence mechanism against potential parasitoids [48]. The phylum Proteobacteria had a preponderant presence in all the samples, but its role in the plant–insect interaction is complex. While some genera have been reported as plant pathogens [124], other Proteobacteria, such as *Pseudomonas spp.*, have been linked to reducing whitefly survival as well as inhibiting aphid growth [125] with potential benefits for the plant. Furthermore, bacteria such as *Serratia symbiotica*, another gamma-proteobacterium, is closely associated with aphids, giving them protection against some parasites [86]. Therefore, it is possible that the phylum Proteobacteria plays different roles in the interaction with both hosts, from mutualistic through commensalisms to antagonistic interactions.

The microbiota of highly embedded systems such as plant–herbivore relationships usually displays multiple kinds of interactions. As mentioned before, network formalism is a suitable tool to study complex biological systems. These intricate relationships could be faithfully represented in multilayer networks. In our case, layers represent profiles of abundances of bacteria in plants and insects. This novel network framework is useful to capture interlayer co-abundance patterns, and the aggregate layer summed up both plant and insect abundance patterns. These patterns of co-occurrence suggest a mixture of microbes from different niches and revealed that Proteobacteria, Actinobacteria, and Firmicutes are ubiquitous, suggesting their potential importance in the functional role of the microbiota in the interaction between the plant *D. inoxia* and its specialist insect *L. daturaphila*.

## 5. Conclusions

Given the nature of the interaction between the annual plant *D. inoxia* and its herbivorous insect *L. daturaphila*, the bacterial communities associated with both hosts shared many phyla and genera, suggesting that the composition of the leaf bacteriome determines the composition of the insect bacteriome. However, the patterns of co-abundances among those bacterial groups differed significantly between plants and insects. That is, those genera that were the most abundant in the plant were not necessarily the most abundant in the insect. Thus, changes in the co-abundance patterns and not in the presence–absence ratio of the bacterial groups conforming the bacteriomes are relevant when studying plant–insect interactions.

## Figures and Tables

**Figure 1 microorganisms-11-00040-f001:**
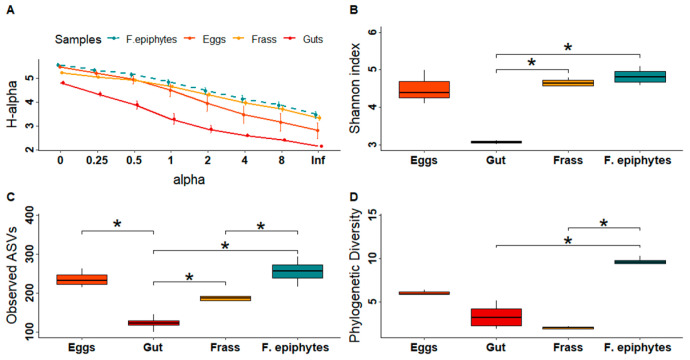
Distribution of alpha diversity in samples collected from the plant *D. inoxia* and its specialist insect *L. daturaphila*. (**A**) Comparison of the Rènyi profiles of bacterial communities considering the total number of ASVs. Alpha values of 0, 1, 2, and infinity correspond to the species richness, Shannon diversity index, the logarithm of the reciprocal Simpson diversity index and Berger–Parker diversity index, respectively. Each colour represents a different sample. Solid lines correspond to insect samples, while dotted lines correspond to plant samples. (**B**) Richness. (**C**) Index of Shannon. (**D**) Phylogenetic diversity. Endophyte and seed samples were not included in this analysis, given the relatively small number of ASVs identified. Asterisks denote significant differences following an ANOVA test (*p* < 0.05).

**Figure 2 microorganisms-11-00040-f002:**
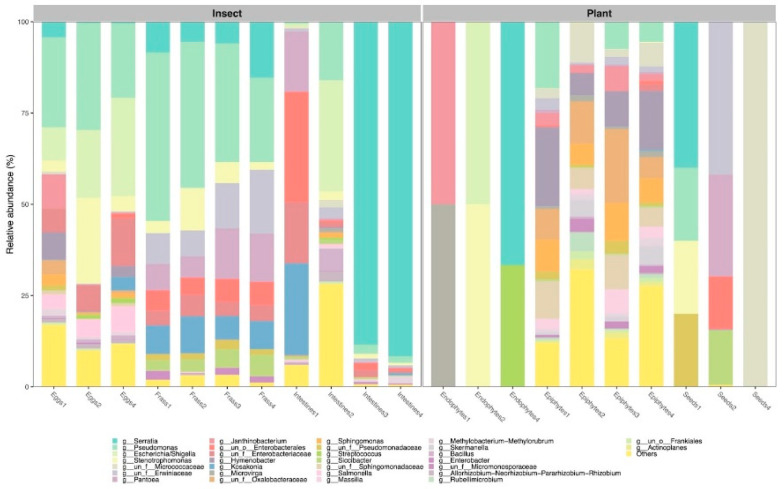
Taxonomic composition of the 12 most abundant bacterial genera of the plant *D. inoxia* and its specialist insect *L. daturaphila.* In the insect, it represented 70.2%, while in the plant, it corresponded to 56.3% of the total readings.

**Figure 3 microorganisms-11-00040-f003:**
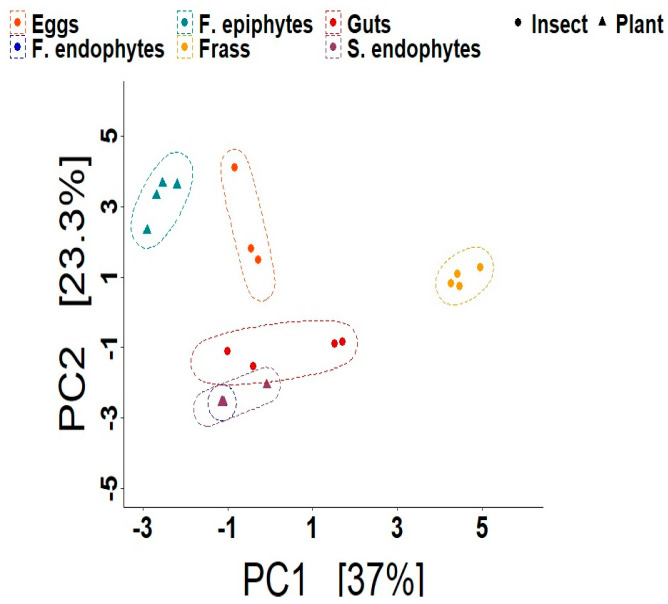
Beta diversity analysis for the samples of *L. daturaphila* (eggs, gut, and frass) and *D. inoxia* (foliar epiphytes). Principal component analysis (PCA) plot based on the Bray–Curtis distance of the ASVs. Each colour represents the bacteriome community in a single type of sample. The symbols represent the origin of the samples: plants or insects. The enclosing ellipses are estimated using the *Khachiyan* algorithm. Endophyte and seed samples were not included in this analysis, given the relatively small number of ASVs identified.

**Figure 4 microorganisms-11-00040-f004:**
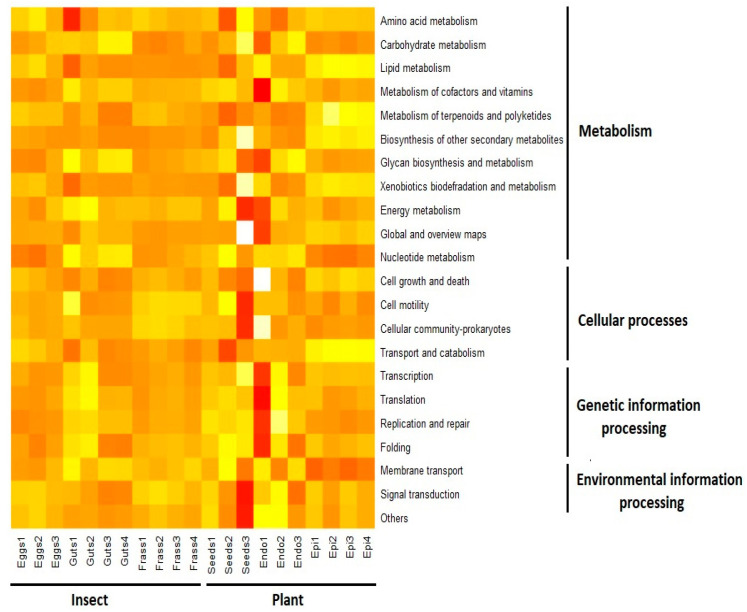
Heatmap of the functional categories inferred from samples collected from the plant *D. inoxia* and its specialist insect *L. daturaphila*. Each column corresponds to a sample, and rows correspond to a specific functional category. The lightest colours represent the highest values, while darker colours correspond to the lowest values.

**Figure 5 microorganisms-11-00040-f005:**
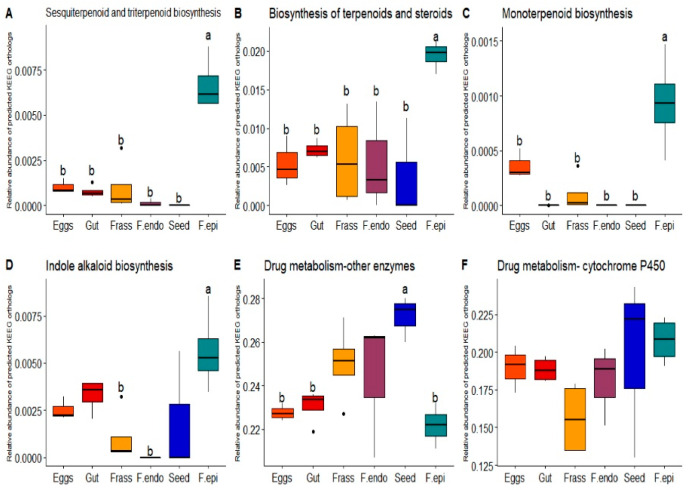
Relative abundances of KEGG orthologs (KOs) at level 1 involved drug metabolism of other enzymes, cytochrome p450, an indole alkaloid, sesquiterpenoid, and terpenoid, monoterpenoid biosynthesis, aminobenzoate degradation, geraniol degradation, and other glycan degradation inferred from samples of the plant *D. inoxia* and its specialist insect *L. daturaphila*. Different letters denote significant differences following a *t*-test (*p* < 0.05).

**Figure 6 microorganisms-11-00040-f006:**
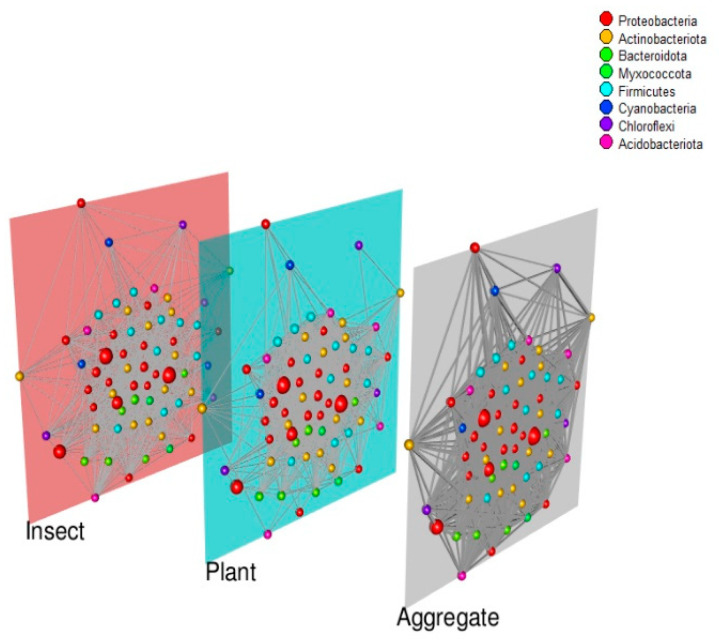
Multilayer network showing the co-abundances among bacterial phyla in the plant *D. inoxia* and its specialist insect *L. daturaphila*. The colours of the nodes correspond to the phylum level, and their sizes are proportional to their abundance in the different samples. Layers depicted the co-abundance networks inferred by the algorithm SPIEC-EASI. The third layer is the resulting aggregate network, where the links are the sum of the weighted individual connections, and the size of the nodes is the sum of the abundances.

## Data Availability

Not applicable.

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
