# Peer review of "Studying Plant–Insect Interactions through the Analyses of the Diversity, Composition, and Functional Inference of Their Bacteriomes"

_microorganisms, 2022, doi:10.3390/microorganisms11010040_

Round 1

Reviewer 1 Report

I would like to thank the authors for their efforts to do this study which describing bacterial community composition of the leaves and seeds Datura inoxia; in addition to bacteriome associated with the eggs, gut, and frass of its specialist insect, Lema daturaphila using DNA extraction followed by the Illumina Miseq platform analysis. However, major concerns should be discussed.

My major concern is the design of experiment. Using only 4 individual plants of Datura from only one site is not enough to study bacterial community composition of epiphytes and endophytes of two plant organs (seeds and leaves). We can not link between bacteriomes of epiphytes and endophytes; consequently, linkage between bacterial community structure and function cannot be concluded.

The main hypothesis of the current study is the bacterial community composition of insect larvae is driven by bacteriome of plant leaves; however, the experimental design which was conducted to address this pion was unclear because DNA was extracted directly from plant organs to study bacteriome structure. So, to link between bacteriomes of plant and insects, DNA of insect eggs and larvae should be immediately extracted from the collected eggs and larvae. Also, transfer larvae collected from the field to under laboratory conditions will certainly change bacterial community associated with insects, even though feeding the larvae with plants collected from the same site.

In order to address the authors hypothesis, different routes should be conducted, and it could be under field or laboratory conditions. For examples, under laboratory condition, larvae feed with different plants. Several factors related to bacteriomes of these plant parts which used should be identified and determined. Under field conditions, many other ways and details should be followed to design that kind of interested study.

In the discussion, section of Co-abundance networks L389, authors presented many details on the used methods with insufficient interpretation of the obtained results.

Using conducted experiment of the current manuscript, we cannot be concluded that “We found that while most phyla and genera were present in both the plant and the insect their abundances and connectivity within the networks were different” L395-396.

Manuscript lacks a conclusion section.

Author Response

Reviewer 1.

I would like to thank the authors for their efforts to do this study which describing bacterial community composition of the leaves and seeds Datura inoxia; in addition to bacteriome associated with the eggs, gut, and frass of its specialist insect, Lema daturaphila using DNA extraction followed by the Illumina Miseq platform analysis. However, major concerns should be discussed.

1. My major concern is the design of experiment. Using only 4 individual plants of Datura from only one site is not enough to study bacterial community composition of epiphytes and endophytes of two plant organs (seeds and leaves). We can not link between bacteriomes of epiphytes and endophytes; consequently, linkage between bacterial community structure and function cannot be concluded.

Response: We thank the reviewer for this comment. We agree that using only four biological replicates does not give us enough information to conclude any general patterns about the composition and function of the bacterial communities. One of the main goals of our study was not to focus on only one of the interacting species but to compare the composition and diversity of the bacterial communities associated with both plants and insects to shed light into their ecological interaction. But, we are aware of the limitations of the study and have thus tone-down the interpretations and conclusions of our study. Please see the revised version of the manuscript (both the Results and Discussion sections).

2. The main hypothesis of the current study is the bacterial community composition of insect larvae is driven by bacteriome of plant leaves; however, the experimental design which was conducted to address this pion was unclear because DNA was extracted directly from plant organs to study bacteriome structure. So, to link between bacteriomes of plant and insects, DNA of insect eggs and larvae should be immediately extracted from the collected eggs and larvae. Also, transfer larvae collected from the field to under laboratory conditions will certainly change bacterial community associated with insects, even though feeding the larvae with plants collected from the same site.

Response: We thank the reviewer for this comment. Indeed, we expected an overlap in the composition of the bacteriomes but differences in the co-abundances. And we also agree with the reviewer in that extracting DNA directly from individuals from the field could help us in linking both bacteriomes. The main reason why we decided to start an insect colony is because, under field conditions, the eggs and larvae can be parasitised. Identifying which eggs or larvae are indeed parasitised is very difficult (you essentially have to wait for the eclosion or emergence). Because the possible presence of the parasitoids in the samples was something we wanted to avoid before DNA extraction, we decided it was better to start a colony. However, we realise that we did not explain this in the manuscript and thus have added this information to better explain our reasoning for this decision. Please see lines 103-109 and lines 127-130.

3. In order to address the authors hypothesis, different routes should be conducted, and it could be under field or laboratory conditions. For examples, under laboratory condition, larvae feed with different plants. Several factors related to bacteriomes of these plant parts which used should be identified and determined. Under field conditions, many other ways and details should be followed to design that kind of interested study.

Response: We totally agree with the reviewer. Different experimental approaches can be followed to address the hypotheses. In the case of our study, we were interested in characterising the bacterial communities associated with both the plant and its herbivorous insect. Given the nature of our study system, and the fact that we could not be certain that the eggs and larvae were not parasitised we decided to collect all the leaves and seeds needed from the field but for the insect samples we decided to start a colony to avoid the presence of the parasitoids in the samples. We are aware of the limitations of our experimental design and have thus tone-down and rewrite the results and discussion of the manuscript. Please see the revised version.

4. In the discussion, section of Co-abundance networks L389, authors presented many details on the used methods with insufficient interpretation of the obtained results.

Response: We thank the reviewer for noticing this and have added more information. Please see lines 184-193 (Data analyses), lines 330-350 (Results) and lines 474-502 (Discussion).

5. Using conducted experiment of the current manuscript, we cannot be concluded that “We found that while most phyla and genera were present in both the plant and the insect their abundances and connectivity within the networks were different” L395-396.

Response: We agree with the reviewer. We have rewrote these lines. Please see lines 469-471.

6. Manuscript lacks a conclusion section.

Response: We thank the reviewer for noticing this. We have added a conclusion section. Please lines 602-610.

Reviewer 2 Report

General comments: -

This study is very interesting and has a scientific topic with a great impact on the field. The manuscript will be suitable for publication after taking care of the following minor comments.

Detailed comments:

1-The English language and /writing style is fine needs some minor check spelling and grammar check

2-Please avoid using the personal pronouns (I, We,) as in line 13 (we characterized)and line 376 (we found) and more.

Abstract

_This section is missing the direct aim of the study. Please state the aim of the study clearly in this section.

_ please state some more values and significant results in this section.

Keywords:

-The keywords has been chosen very carefully and accurately but

- please add the word diversity to the keyword list

Introduction

-The introduction doesn’t provide sufficient background and it is missing enough relevant references

-This section needs to be elongated and enriched with more background about this topic.

Materials and Methods

-it is ok and adequate 

 Results:

The results are very interesting and well presented but some figures needs a better discussion (figure4and 6).

Discussion:

_This section is ok but  , Fig 1 ;Figure 4and Figure 6 needs to be discussed throughly and more clear for better understanding . 

Please rewrite and discuss in details, and fully discussed with related research.

Conclusion 

Please add a conclusion to the current study either in a separate section or as a last paragraph in the discussion.

References

This section is well written and UpToDate. 

Author Response

Reviewer 2

1. General comments: This study is very interesting and has a scientific topic with a great impact on the field. The manuscript will be suitable for publication after taking care of the following minor comments.

Response: We thank the reviewer for the generous comments about our study.

2. Detailed comments: The English language and /writing style is fine needs some minor check spelling and grammar check

Response: We have reviewed the whole manuscript to improve both grammar and style. Please see the revised version of the manuscript.

3. Detailed comments: Please avoid using the personal pronouns (I, We,) as in line 13 (we characterized)and line 376 (we found) and more.

Response: We thank the reviewer for noticing this. We have made all the corrections. Please see the revised version of the manuscript.

 4. Abstract. This section is missing the direct aim of the study. Please state the aim of the study clearly in this section.

Response: We have added the main aim of the study in the abstract. Please see lines 12-18.

5. Abstract. Please state some more values and significant results in this section.

Response: We have added more detailed information about the results in the abstract. Please see lines 18-27.

6. Keywords: The keywords has been chosen very carefully and accurately but please add the word diversity to the keyword list

Response: Done.

7. Introduction. The introduction doesn’t provide sufficient background and it is missing enough relevant references. This section needs to be elongated and enriched with more background about this topic.

Response: We thank the reviewer for noticing this. We have added more information in the Introduction and have included relevant studies and references. Please see lines 36-39, lines 47-65 and lines 70-75.

8. Materials and Methods. -it is ok and adequate 

9. Results: The results are very interesting and well presented but some figures needs a better discussion (figure4and 6).

Response: We agree with the reviewer in that a better explanation of the figures is needed. Please see lines 280-284, lines 295-299 and lines 330-350.

10. Discussion: This section is ok but  , Fig 1 ;Figure 4and Figure 6 needs to be discussed throughly and more clear for better understanding . Please rewrite and discuss in details, and fully discussed with related research.

Response: We agree with the reviewer and have thus added a clearer and better interpretation and discussion of the results. Please see lines 406-423, lines 437-441, 450-460 and lines 475-503.

 11. Conclusion. Please add a conclusion to the current study either in a separate section or as a last paragraph in the discussion.

Response: Done

12. References. This section is well written and UpToDate.

Round 2

Reviewer 1 Report

The revised version was improved. However, as we agreed that “using only four biological replicates does not give us enough information to conclude any general patterns about the composition and function of the bacterial communities.”  So, I recommend the results and discussion section only focus on diversity of the bacterial communities associated with both plants and insects from the collected samples. Please, do not generalize the obtained results, and do not link between bacterial community structure and function based on your experimental design

Author Response

Reviewer 1

Comment: The revised version was improved. However, as we agreed that “using only four biological replicates does not give us enough information to conclude any general patterns about the composition and function of the bacterial communities.”  So, I recommend the results and discussion section only focus on diversity of the bacterial communities associated with both plants and insects from the collected samples. Please, do not generalize the obtained results, and do not link between bacterial community structure and function based on your experimental design.

Response: We have adjusted the Results and Discussion sections to tone-down and limit the extent of the results obtained to our specific study system.